# Subjective Depth & Timescale Transformers: Learning Where and When to Compute

## Abstract

The rigid, uniform allocation of computation in standard Transformer (TF) architectures can limit their efficiency and scalability, particularly for large-scale models and long sequences. Addressing this, we introduce Subjective Depth Transformers (SDT) and Subjective Timescale Transformers (STT), two distinct architectures that leverage Bayesian surprise signals to dynamically route computation, learning where and when to compute within decoder-only TFs. SDT augments a decoder-only stack with alternating Decision and Dynamic layers: a Decision layer computes a full block 'posterior' and a lightweight 'prior,' while a Dynamic layer employs fixed-capacity Top-K routing based on Bayesian surprise (Expected and Unexpected Change), maintaining a static compute graph. STT extends this conditional computation to the temporal domain: a transition network predicts residual updates, forming a temporal 'change hypothesis' that informs a router to dynamically execute or bypass TF blocks for each token, managing KV-cache contributions. Both architectures exhibit the predicted shift from novelty to prediction driven gating over training, suggesting alignment with surprise based principles. While operating at reduced capacity, they offer preliminary insights into the compute-accuracy trade-offs of conditional computation. The proposed architectures establish a flexible framework for efficiency, reducing self-attention computation by 75% and KV-cache requirements by 50% within each compute skipping layer, setting a pathway for more efficient models.

## 1 Introduction

The uniform allocation of computation across tokens and layers in Transformer (TF) architectures presents challenges to their efficiency and scalability (Raposo et al., 2024). While this design has established TFs as the foundational architecture for modern large language models (LLMs) (Vaswani et al., 2017; Minaee et al., 2025), its inefficiency becomes particularly pronounced when processing long sequences, as the self-attention mechanism's computational cost scales quadratically with sequence length ($O(T^2)$) (Dao et al., 2022). This rigid expenditure is often misaligned with the non-uniform distribution of information in language, as not all tokens require the same degree of processing to make an accurate prediction (Raposo et al., 2024). To address this, a significant body of work has explored conditional computation, with methods like Mixture-of-Experts (MoE) (Shazeer et al., 2017; Fedus et al., 2022) and Mixture-of-Depths (MoD) (Raposo et al., 2024) learning to route tokens to specialised or optional computational paths. Building on this paradigm, we introduce two novel architectures, the Subjective Depth Transformer (SDT) and the Subjective Timescale Transformer (STT), which leverage a surprise-based mechanism to dynamically allocate computation.

**Conditional Compute Challenges.** A key challenge for dynamic computation is enabling token-level routing while preserving the static computation graphs and predictable tensor shapes favoured by modern hardware accelerators (Raposo et al., 2024), (Paszke et al., 2019). The MoD architecture provides a solution to this constraint (Raposo et al., 2024). At designated layers, a small learnable router assigns a scalar score to each token. An expert-choice routing scheme then selects a fixed-capacity subset of tokens with the highest scores-the Top-K-for processing by the full TF block. All remaining tokens bypass this computationally intensive path via a residual connection (Raposo et al., 2024). Because the number of selected tokens is defined *a priori*, this mechanism maintains a static computation graph, ensuring hardware efficiency (Raposo et al., 2024). However, the Top-K

operation is inherently non-causal, as a token's inclusion depends on the scores of all subsequent tokens. To enable autoregressive inference, MoD therefore trains a separate, lightweight causal predictor to approximate the non-causal routing decisions at generation time (Raposo et al., 2024).

**Bayesian Surprise Principle.** In contrast to MoD, we propose an alternative for conditional computation grounded in the information-theoretic concept of Bayesian surprise (Itti & Baldi, 2005; MacKay, 2003). Bayesian surprise quantifies the degree of belief update in an observer's internal model upon receiving new data, defined as the Kullback-Leibler (KL) divergence between the posterior $P(M|D)$ and prior $P(M)$ belief distributions over a model class $\mathcal{M}$ (Itti & Baldi, 2005):

$$S(D, \mathcal{M}) = KL(P(M|D), P(M)) = \int_{\mathcal{M}} P(M|D) \log \frac{P(M|D)}{P(M)} dM \tag{1}$$

The principle is motivated by findings in cognitive science, where surprising events are shown to attract human attention and trigger the segmentation of continuous experience into meaningful episodes (Itti & Baldi, 2005; Fountas et al., 2022; Kumar et al., 2023; Fountas et al., 2025). This concept has been successfully operationalised in hierarchical variational models for video encoding and prediction, such as Variational Predictive Routing (VPR) and Dynamic Latent Hierarchy (DLH), where surprise-based signals guide the dynamic gating of information flow across temporal and spatial scales (Zakharov et al., 2022; 2023). We hypothesise that by applying this mechanism a TF can learn to allocate computation only to tokens inducing a significant update in its internal representations, providing a more effective inductive bias for routing than a generic importance score.

**Approximations of Bayesian Surprise.** To use the idea of Bayesian surprise within a standard TF, which lacks explicit probabilistic parameters, we introduce an approximation to the KL. We treat the token hidden state vectors as the means of underlying isotropic Gaussian distributions with a shared covariance ($\Sigma = kI$). Under this assumption, the general KL divergence formula between a posterior $p(\cdot) \sim \mathcal{N}(\mu_p, kI)$ and a prior $q(\cdot) \sim \mathcal{N}(\mu_q, kI)$ simplifies significantly (MacKay, 2003). This yields a divergence proportional to the squared Euclidean distance between the mean vectors:

$$D_{KL}(\mathcal{N}(\mu_p, kI)||\mathcal{N}(\mu_q, kI)) \propto ||\mu_p - \mu_q||_2^2 \tag{2}$$

This result provides a justification for using the Mean Squared Error (MSE) between hidden state vectors as a tractable proxy for Bayesian surprise, forming the core of our routing mechanisms. More details on how to arrive at this approximation are provided in the Appendix A.

**Subjective Depth Transformers (SDTs).** The SDT architecture implements surprise-based routing by modifying a standard decoder-only stack into a sequence of alternating Decision Layers and Dynamic Layers. Each decision layer processes its input through a standard TF block to produce a posterior state. In parallel, a computationally inexpensive Prior Feed-Forward Network (PriorFFN) generates a prior state by predicting the output of the main block (Raposo et al., 2024; Zakharov et al., 2022). The subsequent Dynamic Layer receives the original, prior, and posterior states and employs a Predictive Router, inspired by VPR (Zakharov et al., 2022), to compute token-wise surprise scores. These scores are derived by comparing the static hypothesis (posterior vs. original) against the change hypothesis (posterior vs. prior). Finally, the router selects a fixed-capacity subset of tokens using a Top-K mechanism, analogous to MoD (Raposo et al., 2024), and only these selected tokens are processed by the Dynamic Layer's own TF block, while the rest bypass it.

**Subjective Timescale Transformers (STTs).** The STT adapts the surprise-based routing principle to the temporal domain. Instead of comparing a block's output to a parallel prior network's prediction for the same token, STT predicts the residual change for the current token, $t$, based on the processed state of the previous token, $t - 1$. This prediction is generated by a lightweight Transition Network (TPN). The resulting temporal change hypothesis is then compared against the actual residual produced by the full TF block, forming a surprise signal that is more directly aligned with the event-detection mechanisms in sequential models like VPR (Zakharov et al., 2022). This approach simplifies the architecture by integrating the signal generation and conditional execution into a single unified layer, removing the need for alternating Decision and Dynamic layers. The intuition is that the causal self-attention mechanism ensures the representation of the previous token is a powerful and efficient predictor for the current token's state update (Vaswani et al., 2017).

**Contributions.** In this work, we introduce a surprise-driven framework for conditional computation in decoder-only TFs. Our primary contributions are:

- The design and implementation of two novel conditional compute architectures, SDT and STT, each integrating a routing mechanism based on Bayesian surprise to allocate computation.
- A empirical comparison of our proposed architectures against a re-implemented MoD baseline under a fixed-capacity, transfer-learning regime. This provides a controlled analysis of the compute-accuracy trade-offs between heuristic and surprise-driven routing.
- An analysis of the internal routing dynamics, demonstrating that our models learn a policy consistent with the principles of predictive coding showcased in the results, Section 4.

## 2 BACKGROUND

### 2.1 ROUTING VIA BAYESIAN SURPRISE

Our work is principally inspired by models utilising theories of predictive coding from neuroscience (Rao & Ballard, 1999; Friston, 2010). This framework posits that the brain minimises prediction error by comparing internal predictions against sensory data. A key metric in this process is Bayesian surprise, which quantifies the degree of belief update upon observing new data (Itti & Baldi, 2005).

This principle has been successfully instantiated in hierarchical models for video prediction, such as VPR and the DLH (Zakharov et al., 2022; 2023). These models employ an event-detection mechanism that evaluates two competing hypotheses at each timestep: a static hypothesis that the latent state remains unchanged, and a change hypothesis that it evolves according to a learned transition model. The decision to update a latent state is framed as a model comparison, governed by which hypothesis results in lower surprise, measured via the KL divergence. VPR defines two specific criteria for this decision, here, $D_{st} = D_{KL}(q_{st}||p_{st})$ quantifies the surprise from a new observation under the static hypothesis, whereas $D_{ch} = D_{KL}(q_{ch}||p_{ch})$ quantifies it under the change hypothesis:

- **Expected Change (CE):** An event is detected if the change hypothesis is a better explanation for the new data than the static one, i.e., $D_{st} > D_{ch}$.
- **Unexpected Change (CU):** An event is detected if the surprise under the static hypothesis, $D_{st}$, significantly exceeds its recent moving average, i.e., $D_{st} > \gamma \cdot \text{MA}(D_{st})$.

### 2.2 DECODER-ONLY TRANSFORMERS

The modern decoder-only TF is an autoregressive language model that factorises the joint probability of a token sequence $x_{1:T}$ using the chain rule, $p_\theta(x_{1:T}) = \prod_{t=1}^{T} p_\theta(x_t|x_{<t})$ (Vaswani et al., 2017). The model is trained by minimising the negative log-likelihood of predicting the next token given its causal context. The architecture consists of a stack of identical blocks/layers, each containing two primary sub-layers: multi-head self-attention and a position-wise feed-forward network (FFN), typically a SwiGLU variant (Shazeer, 2020). Both sub-layers are wrapped with residual connections and layer normalisation, commonly pre-normalisation using Root Mean Square Layer Normalisation (RMSNorm) (Zhang & Sennrich, 2019).

For an input sequence representation $X \in \mathbb{R}^{T \times d}$, where $T$ is the sequence length and $d$ is the model's hidden dimension, linear projections yield queries ($Q$), keys ($K$), and values ($V$). The scaled dot-product attention is a weighted sum of these values:

$$\text{Attention}(Q, K, V) = \text{softmax}\left(\frac{QK^\top}{\sqrt{d_k}} + M\right) V \tag{3}$$

where $d_k$ is the key dimension and $M$ is a causal mask to prevent attention to future positions. To handle long sequences efficiently during inference, implementations rely on a key-value (KV) cache, which stores the keys and values of past tokens for reuse in subsequent generation steps.

**Conditional Computation in TFs** The computational cost and uniform processing of dense TFs have motivated a range of conditional computation methods designed to improve efficiency. The Universal Transformer employs a single, recurrent block with tied weights and a dynamic halting

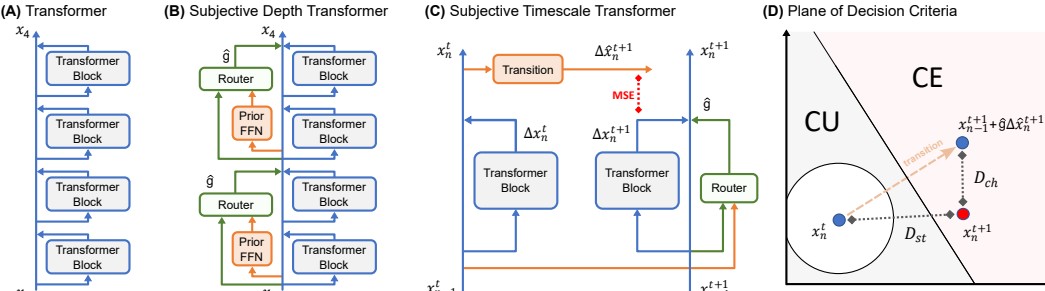

Figure 1: Architectural comparison and routing criteria visualisation. **(A)** A standard decoder-only TF processes all tokens through every block. **(B)** SDT augments the standard stack with alternating Decision and Dynamic layers. At Decision layers, a lightweight PriorFFN generates a prediction of the main block's output, and both are passed to a Router. **(C)** STT uses a Transition Network to form a temporal prediction for the current token $t$ based on the previous token's state $t-1$. **(D)** Plane of decision criteria, providing a geometric intuition for the routing logic. From a starting point or static prior ($x_n^t$), a transition network provides a change prior. The distance from each of these priors to the true posterior ($x_n^{t+1}$) is measured as static surprise ($D_{st}$) and change surprise ($D_{ch}$), respectively. A token is processed if the change prior is a better explanation for the posterior (CE) or if the static surprise is unusually large (CU).

mechanism, allowing each token to undergo a variable number of processing steps which leads to variable computation graphs(Dehghani et al., 2018; Graves, 2016). Conversely, our architectures, like MoD, allow tokens to bypass one layer but re-enter computation at a subsequent one, and our fixed-capacity design ensures predictable hardware utilisation.

**MoE and MoD.** MoE layers replace the dense FFN with a set of $E$ parallel FFNs and a learnable router (Shazeer et al., 2017; Fedus et al., 2022; Jiang et al., 2024). For each token $x_t$, the router computes logits $l(x_t) = x_t W_g$ and selects the Top-K experts. The final output is a weighted combination of the outputs from the selected experts. By activating only a small subset of experts ($K \ll E$) for each token. MoD applies a similar routing concept to TF blocks (Raposo et al., 2024). At a given layer $l$, a lightweight router computes a scalar score $r_t^{(l)}$ for each token $x_t^{(l-1)}$. A Top-K mechanism selects the set of indices $\mathcal{S}^{(l)}$ corresponding to the $k = \lfloor \gamma T \rfloor$ tokens with the highest scores to be processed by the full TF block $f^{(l)}$; the rest bypass it via a residual connection. The update is performed by scattering the results back to their original positions:

$$x_t^{(l)} = \begin{cases} r_t^{(l)} \cdot f^{(l)}(X_{\mathcal{S}^{(l)}}^{(l-1)})_t + x_t^{(l-1)} & \text{if } t \in \mathcal{S}^{(l)} \\ x_t^{(l-1)} & \text{if } t \notin \mathcal{S}^{(l)} \end{cases} \tag{4}$$

Notably, the output of the function $f^{(l)}$ is multiplied by the router weights, which subjects them to gradient descent during training. This fixed-capacity design preserves a static computation graph. However, since the Top-K selection is non-causal, MoD requires a separate, causally-constrained predictor for autoregressive inference (Raposo et al., 2024). While our work adopts fixed-capacity routing, we replace MoD's heuristic strategy with a mechanism grounded in Bayesian surprise.

## 3 METHODOLOGY

We introduce two novel architectures for conditional computation in decoder-only TFs, the SDT and STT, designed to be integrated into a pre-trained TF to replace a subset of standard decoder layers.

### 3.1 SUBJECTIVE DEPTH TRANSFORMER (SDT)

The SDT architecture (Figure 1B) implements surprise-based routing by separating the generation of predictive signals from the execution of conditional computation. It modifies a standard decoder-only stack (Figure 1A) into a sequence of alternating Decision Layers and Dynamic Layers.

### 3.1.1 THE DECISION LAYER

The Decision Layer is designed to generate, in parallel, the three representations required for the subsequent surprise calculation. For an input token representation $x_t^{(l-1)}$ at layer $l$, the layer computes the true block residual $\Delta x_t^{(l)}$ and a predicted residual $\widehat{\Delta x}_t^{(l)}$:

$$x_{t,post}^{(l)} = x_t^{(l-1)} + \text{TF-Block}(x_t^{(l-1)}) \tag{5}$$

$$\Delta x_t^{(l)} = x_{t,post}^{(l)} - x_t^{(l-1)} \tag{6}$$

$$\widehat{\Delta x}_t^{(l)} = \text{PriorFFN}(x_t^{(l-1)}) \tag{7}$$

The PriorFFN is a computationally inexpensive MLP trained with an auxiliary MSE loss to approximate the full block's transformation: $\mathcal{L}_{prior} = \text{MSE}(\widehat{\Delta x}_t^{(l)}, \text{stop\_gradient}(\Delta x_t^{(l)}))$ We parametrise the PriorFFN's intermediate width by a *prior factor* $f$, setting $d_{\text{inter}} = \lceil f \cdot d \rceil$, which allows explicit control over the predictor's capacity versus its computational cost.

### 3.1.2 THE DYNAMIC LAYER

The subsequent Dynamic Layer receives the actual and predicted residuals from the Decision Layer to perform conditional computation. It first computes the token-wise surprise metrics:

$$D_{st,t}^{(l)} = \frac{1}{d}||\Delta x_t^{(l)}||_2^2 \tag{8}$$

$$D_{ch,t}^{(l)} = \frac{1}{d}||\Delta x_t^{(l)} - \widehat{\Delta x}_t^{(l)}||_2^2 \tag{9}$$

These metrics are fed into the unified surprise routing mechanism (Section 3.3) to produce a gating score for each token. A Top-K selection based on these scores identifies the tokens to be processed by the Dynamic Layer's own TF block.

## 3.2 SUBJECTIVE TIMESCALE TRANSFORMER (STT)

The STT (Figure 1C) adapts the surprise principle to the temporal domain, simplifying the architecture by integrating signal generation and execution into a single, unified STT Layer. Instead of a spatial prior, the STT uses a lightweight Transition Network (TPN) to predict the residual temporal change for the current token $t$ using the processed state of the *previous* token, $t-1$.

$$\widehat{\Delta x}_t^{(l)} = \text{TPN}^{(l)}(x_{t-1}^{(l)}) \tag{10}$$

The surprise metrics are then calculated by comparing this temporal prediction to the actual residual, $\Delta x_t^{(l)} = x_t^{(l)} - x_t^{(l-1)}$, where $x_t^{(l)}$ is the output of the STT Layer's internal TF block.

$$D_{st,t}^{(l)} = \frac{1}{d}||\Delta x_t^{(l)}||_2^2 \tag{11}$$

$$D_{ch,t}^{(l)} = \frac{1}{d}||\Delta x_t^{(l)} - \widehat{\Delta x}_t^{(l)}||_2^2 \tag{12}$$

These metrics are then used by the unified routing mechanism to determine whether to execute the internal TF block for token $t$. In addition to the default fixed-capacity routing via Top-K, we also experiment with a variable-capacity version for the STT that uses a threshold on the gating scores. This allows the model to adjust the number of processed tokens per layer based on the complexity of the input sequence; this means we no longer have a static computational graph, which can limit efficiency with current hardware (Raposo et al., 2024).

## 3.3 UNIFIED SURPRISE ROUTING AND TRAINING

Both SDT and STT employ the same core routing mechanism to convert surprise metrics into a differentiable, fixed-capacity gating decision. This logic is visualised in Figure 1D.

### 3.3.1 DIFFERENTIABLE GATING CRITERIA

The two event criteria from VPR are reformulated as continuous, differentiable signals by using subtraction instead of a hard inequality, allowing gradients to flow through the routing decision. The criteria are modulated by learnable parameters: prediction offset $o_{ce}$ and novelty multiplier $m_{cu}$.

$$\text{CE}_t = D_{st,t} - (D_{ch,t} - \log(o_{ce})) \tag{13}$$

$$\text{CU}_t = D_{st,t} - (m_{cu} \cdot \text{MA}(D_{st,t})) \tag{14}$$

Here, $\text{MA}(\cdot)$ denotes a moving average over the token sequence. These signals are converted to probabilities using a scaled sigmoid, where the inverse temperatures $\beta_{ce}$ and $\beta_{cu}$ are learnable and annealed over training. The final continuous gating score, $g_{cont,t} \in [0,1]$, is computed using a probabilistic OR:

$$g_{cont,t} = \sigma(\beta_{ce} \cdot \text{CE}_t) + \sigma(\beta_{cu} \cdot \text{CU}_t) - \sigma(\beta_{ce} \cdot \text{CE}_t)\sigma(\beta_{cu} \cdot \text{CU}_t) \tag{15}$$

The conditionally executed TF block output is then scaled by $g_{cont,t} \in [0,1]$ before the residual connection, like MoD, allowing gating values to be directly shaped by language modelling objectives. Under fixed-capacity Top-K routing, the continuous gating score $g_{cont,t}$ serves directly as the token importance score in MoD, yielding precisely the same selection semantics when applying Top-K.

### 3.3.2 TRAINING OBJECTIVES AND INFERENCE

The total training objective combines the primary language modeling loss ($\mathcal{L}_{LM}$) with several weighted auxiliary losses designed to guide the routing mechanism. The final loss is defined as:

$$\mathcal{L}_{total} = \mathcal{L}_{LM} + \lambda_{pred}\mathcal{L}_{pred} + \lambda_{causal}\mathcal{L}_{causal} + \lambda_{g\_reg}\mathcal{L}_{g\_reg}$$

Here, $\mathcal{L}_{pred}$ is a MSE loss that trains the predictive network (the PriorFFN in SDT or the TPN in STT) to forecast the true block residual, weighted by $\lambda_{pred} = 0.05$. $\mathcal{L}_{causal}$ is a Binary Cross-Entropy loss that trains the Causal Router (CR) to mimic the Non-Causal Router's decisions, weighted by $\lambda_{causal} = 0.01$. Finally, $\mathcal{L}_{g\_reg}$ is an optional loss, primarily for the variable-capacity STT, that regularises the continuous gating scores to encourage sparsity, weighted by $\lambda_{g\_reg} = 0.001$.

During training, the non-causal gating score, $g_{cont}$, is used to generate a binary target mask via Top-K or thresholding. A lightweight CR is trained to predict this mask using only causally available inputs. The SDT's CR is an MLP that takes the token's input state $x_t^{(l-1)}$ as input, analogous to the design in MoD (Raposo et al., 2024). The STT's CR takes the concatenated input states of the current and previous tokens, $[x_t^{(l-1)} \| x_{t-1}^{(l-1)}]$, to better leverage temporal context. At inference time, only this efficient CR is active, ensuring that the routing decision is strictly autoregressive.

### 3.3.3 TOKEN SELECTION MECHANISMS: TOP-K VS. THRESHOLD

**Fixed-Capacity (Top-K) Routing.** This is the default mechanism for both SDT and STT, inspired by MoD (Raposo et al., 2024). For a given sequence of length $T$, it selects the $k = \lfloor \gamma T \rfloor$ tokens with the highest gating scores, where the capacity $\gamma$ is a fixed hyperparameter. The primary advantage of this approach is that it guarantees a constant number of tokens are processed, thereby preserving a static computation graph.

**Variable-Capacity (Threshold) Routing.** As an alternative explored with the STT, we use a simple thresholding mechanism. A token is selected for processing if its gating score exceeds a predefined hyperparameter, $g_{cont,t} \geq g_{th}$. This approach allows the model to adjust its computational budget on a per-layer, per-sequence basis, potentially processing fewer tokens for simpler inputs and more for complex ones. However, this flexibility comes at the cost of a dynamic computation graph, where the number of selected tokens can vary, which may lead to less efficient hardware utilisation compared to the fixed-capacity approach.

## 4 EXPERIMENTS AND RESULTS

### 4.1 EXPERIMENTAL SETUP

Our experiments are performed in a transfer-learning regime. We adapt a pre-trained Qwen2.5-0.5B model, a decoder-only TF to serve as the backbone for all architectures (Team et al., 2025).

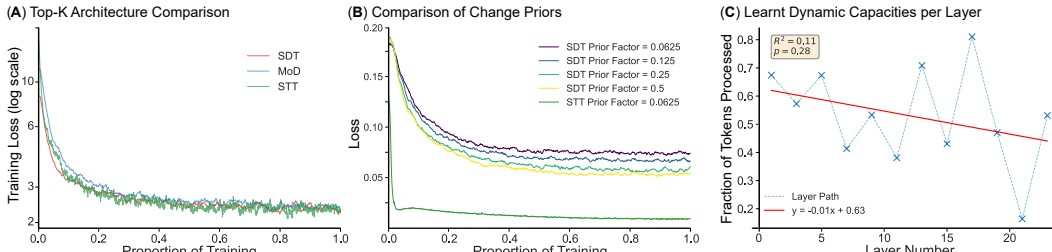

Figure 2: (**A**) Training loss for the fixed-capacity SDT, STT, and MoD models at $\gamma = 0.5$. Curves are smoothed with an exponential moving average over the last 15 steps. (**B**) Comparison of the predictive loss ($\mathcal{L}_{pred}$) for the STT's TPN and the SDT's PriorFFN with varying intermediate size factors. The temporal prior (STT) is consistently more accurate. (**C**) The average proportion of tokens selected per layer by the STT (Dynamic Capacity) model during validation. The model learns to reduce its computational capacity in deeper layers.

Weights from the pre-trained model were used to initialise the corresponding TF blocks in our architectures. All new parameters, such as those in the PriorFFN and TPN, were initialised from a normal distribution $\mathcal{N}(0, 0.02^2)$, while the learnable biases in the Predictive Router were initialised to values informed by hyperparameter sweeps. All models were fine-tuned on the same mixed-domain corpus using the AdamW optimiser (Kingma & Ba, 2014; Loshchilov & Hutter, 2017).

We compare a Dense Baseline (The standard, unmodified Qwen2.5-0.5B model) (Team et al., 2025), a MoD Baseline (Our re-implementation of MoD architecture)(Raposo et al., 2024), a SDT (Fixed Capacity), a STT (Fixed Capacity), and a STT (Dynamic Capacity) an STT variant using a threshold-based router instead of Top-K, allowing it to learn a per-layer processing capacity. In all model variants, the cost-saving layer is interleaved with standard TF blocks (i.e., applied only every other layer), aligning with the SDT design and the training stability benefits reported by Raposo et al. (2024).

Unless specified otherwise, all models with fixed capacity were configured with $\gamma = 0.5$. Model performance was assessed using a suite of downstream benchmarks: MMLU (Hendrycks et al., 2021), ARC-Challenge (Clark et al., 2018), HellaSwag (Zellers et al., 2019), WinoGrande (Sakaguchi et al., 2021), and TruthfulQA (Lin et al., 2021).

## 4.2 EXPERIMENTAL RESULTS

**Comparative Analysis of Architectures** All three architectures exhibit stable training profiles (Figure 2A), with training loss decreasing at a comparable rate, demonstrating that surprise-based routing is as effective as the heuristic-based approach of MoD in this regime. The downstream benchmark results are presented in Table 1. While no single architecture consistently outperforms the others across all tasks, the surprise-based models, particularly the STT variants, achieve the highest scores on several benchmarks. Notably, the STT with dynamic capacity, which learns to allocate its computational budget, achieves competitive results relative to the other conditional models. As expected in this limited transfer-learning setting, all conditional compute models underperform the dense baseline, highlighting the trade-off between efficiency and accuracy.

The STT (Dynamic Capacity) variant provides insight into how a surprise-based model learns to structure its computation. Figure 2C shows the average proportion of tokens selected for processing at each STT Layer, inferred during validation. The model learns to decrease its computational allocation in deeper layers, a behaviour consistent with the nested timescale structure observed in hierarchical models like VPR and DLH (Zakharov et al., 2022; 2023). This suggests that the model allocates more resources to lower-level features processed in earlier layers, while using less compute for the more abstract, slower-changing representations hypothesised to be processed in deeper layers.

**Causal Router Performance.** A key component for efficient autoregressive inference is the lightweight Causal Router (CR), trained to mimic the non-causal Top-K decisions. Our experiments revealed a notable difference between architectures. The STT's CR, which conditions on both

Table 1: Benchmark performance for 0.5B parameter models. Scores are accuracy (%).

| Model Variant | ARC-C | HellaSwag | MMLU | TruthfulQA | WinoGrande |
|---|---|---|---|---|---|
| **Number of Shots** | 25 | 10 | 5 | 0 | 5 |
| Dense Baseline | 43.7 | 52.1 | 55.9 | 40.2 | 56.3 |
| MoD Baseline | 24.3 | 32.6 | 23.3 | 43.5 | **52.6** |
| SDT (Fixed Capacity) | 25.9 | **33.3** | 24.4 | 44.5 | 50.7 |
| STT (Fixed Capacity) | 25.9 | 26.3 | 26.5 | **48.2** | 52.1 |
| STT (Dynamic Capacity) | **27.2** | 26.5 | **26.6** | 46.8 | 50.3 |
| Random Guessing | ∼25 | 25 | 25 | ∼23 | 50 |

the current ($x_t^{(l-1)}$) and previous ($x_{t-1}^{(l-1)}$) token states, was effective at predicting the gating mask. In contrast, the SDT's CR, which only uses the current token state ($x_t^{(l-1)}$), struggled to learn this mapping effectively. This suggests that the temporal context available to the STT's router is crucial for making accurate causal predictions.

**Compute and Memory Savings.** Our architectures, with custom layers interleaved with standard layers, offer significant savings. For the fixed-capacity models at $\gamma = 0.5$, the self-attention workload is reduced to 62.5% of the dense baseline, yielding a 37.5% saving. This is because half the layers operate at full cost while the other half operate at $0.5^2 = 25\%$ of the self-attention cost. For the STT (Dynamic Capacity) variant, which learns an average processing capacity $\bar{\gamma}$ per layer, the total self-attention savings are $(1 - \bar{\gamma}^2)/2$, which for this experiment was 35.94%. These routing schemes also reduce the KV-cache size, this results in a memory saving of $(1 - \bar{\gamma})/2$ relative to the dense baseline, roughly 25%. Notably, these savings could be nearly doubled if the STT architecture were applied to every layer instead of every other layer, though we used the latter approach to ensure a fair comparison with our SDT design and the stable configuration reported in MoD Raposo et al. (2024).

**Ablation Studies** We conducted ablation studies to analyse the impact of the predictive prior's design and the fine-tuning strategy on performance. Figure 2B compares the predictive loss ($\mathcal{L}_{pred}$) for the SDT's PriorFFN across different intermediate sizes and the STT's TPN. The temporal prior of the STT is significantly more accurate than the spatial prior of the SDT, achieving a much lower prediction error, suggesting a previous token's state is a more effective predictor of the current token's residual than the current token's own input state.

Table 2 presents the downstream benchmark performance for these ablations. For SDT, increasing the PriorFFN's expressivity does not uniformly improve performance, indicating that a lightweight prior is sufficient to generate an effective surprise signal. The comparison between full fine-tuning and a parameter-efficient approach using LoRA (Hu et al., 2021) on the SDT's base TF blocks shows full fine-tuning yields better performance on several common-sense benchmarks, suggesting that fully adapting the base components is beneficial, though LoRA remains a possible alternative.

Table 2: Ablation study results for 0.5B SDT models. All operate at a fixed capacity of $\gamma = 0.5$.

| Model Variant | ARC-C | HellaSwag | MMLU | TruthfulQA | WinoGrande |
|---|---|---|---|---|---|
| *Prior Expressivity Ablation* | | | | | |
| SDT (Prior=0.0625) | 25.9 | **33.3** | **24.4** | 44.5 | 50.7 |
| SDT (Prior=0.125) | 26.0 | 32.9 | 24.1 | 44.5 | **51.9** |
| SDT (Prior=0.25) | 25.3 | 33.2 | 23.6 | **45.6** | 48.8 |
| SDT (Prior=0.5) | **26.3** | 33.2 | 24.1 | 45.1 | 50.4 |
| *Adaptation Strategy Ablation* | | | | | |
| SDT (Finetune) | 25.9 | **33.3** | **24.4** | 44.5 | **50.7** |
| SDT (LoRA) | **27.2** | 26.6 | 24.1 | **48.3** | 48.7 |

## 5 DISCUSSION

**Agreement with Predictive-Coding Theory**    In this work we translated the surprise-based gating from models like VPR into a TF setting (Zakharov et al., 2022). Our results show several points of qualitative agreement with the principles of predictive coding. Both the SDT and STT architectures exhibit a consistent shift in their routing dynamics over the course of training: the novelty-driven CU signal is more influential initially, while the prediction-driven CE signal becomes dominant as the predictive networks (PriorFFN and TPN) become more accurate. This mirrors the dynamics observed in VPR, where the CU criterion serves to bootstrap learning before a reliable predictive model is formed (Zakharov et al., 2022).

Furthermore, the STT variant provides empirical support for nested timescales in hierarchical processing. The model learns to follow a trend to reduce its computational capacity in deeper layers (Figure 2C), suggesting it allocates more resources to the local, faster-changing features processed in earlier layers, and fewer to the more abstract, slower-changing representations hypothesised to exist in deeper layers (Zakharov et al., 2022; 2023). Together, these findings suggest that surprise-based gating is a viable and theoretically grounded inductive bias for conditional computation in TFs.

**Compute-Accuracy Trade-offs and Baselines**    All models underperformed the dense baseline on downstream benchmarks (Table 1). This is an expected outcome in a transfer-learning setting where the total per-token computation is significantly reduced without the benefit of large-scale pre-training to compensate. The goal of this study was not to claim superior absolute performance, but to compare the relative efficacy of different routing criteria under a matched computational budget. Our ablation on the SDT's PriorFFN expressivity further revealed that a lightweight prior is sufficient to generate an effective surprise signal, which is encouraging for designing efficient architectures.

**Limitations and Future Directions**    Our results provide a controlled comparison under a specific transfer-learning regime, designed to isolate the efficacy of the routing mechanism (Fedus et al., 2022; Raposo et al., 2024). Future work could conduct comprehensive capacity sweeps (e.g., for $\gamma \in \{0.25, 0.5, 0.75\}$) and scale the experiments to larger models and longer training schedules to fully map the compute-accuracy trade-offs. Improving and rigorously evaluating the causal router for the SDT architecture is a key direction for realising practical latency improvements in autoregressive generation. Further work could also explore more calibrated surprise metrics beyond MSE, such as layer-normalised cosine similarity, and investigate depth-wise capacity schedules to more explicitly model hierarchical processing. Finally, providing robust statistical evidence by running experiments across multiple seeds is needed to validate the significance of the observed performance differences.

## 6 CONCLUSION

This work introduced the SDT and STT, two novel conditional compute architectures that allocate resources selectively across depth and time. Our models replace the heuristic-based routers of prior work (Raposo et al., 2024) with a principled gating mechanism grounded in Bayesian surprise. Our empirical findings demonstrate that these architectures train stably and exhibit routing dynamics consistent with their theoretical underpinnings, in this setting at least. However, the results also highlight the clear compute-accuracy trade-off inherent in conditional computation, as all variants underperform the dense baseline on downstream benchmarks within our limited adaptation regime. Our ablations further revealed that a lightweight predictive network is sufficient to generate an effective surprise signal for the SDT, and that a full fine-tuning of the TF blocks yields better performance than a more parameter-efficient adaptation using LoRA. The scope of these conclusions is constrained by several limitations, including the use of a single model scale, a single capacity setting, and the lack of statistical testing across multiple seeds.

In summary, this work establishes surprise-based routing as an effective alternative for conditional computation in Transformers and laying the groundwork for future investigation into its scaling properties and practical benefits.

## 7 REPRODUCIBILITY STATEMENT

We specify all implementation and training details in the Appendix B: data processing steps; model architectures; learning rates; batch sizes; optimiser configurations; training schedules; random seed choices.

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

## A   KL Divergence Approximation

### Proposition

Let $p(\mathbf{z})$ and $q(\mathbf{z})$ be two multivariate Gaussian distributions in $\mathbb{R}^d$, defined as $p(\mathbf{z}) = \mathcal{N}(\mathbf{z}|\boldsymbol{\mu}_p, k\mathbf{I})$ and $q(\mathbf{z}) = \mathcal{N}(\mathbf{z}|\boldsymbol{\mu}_q, k\mathbf{I})$, where $k \in \mathbb{R}^+$ is a positive scalar variance and $\mathbf{I}$ is the identity matrix. The Kullback-Leibler (KL) divergence from $q$ to $p$ is equivalent to the squared Euclidean distance between their means, scaled by the inverse of twice their variance.

$$D_{KL}(p||q) = \frac{1}{2k}||\boldsymbol{\mu}_p - \boldsymbol{\mu}_q||_2^2$$

### Proof

The Kullback-Leibler divergence between two continuous probability distributions $p(\mathbf{z})$ and $q(\mathbf{z})$ is defined as:

$$D_{KL}(p||q) = \int p(\mathbf{z}) \log \frac{p(\mathbf{z})}{q(\mathbf{z})} d\mathbf{z}$$

For two multivariate Gaussian distributions, $p = \mathcal{N}(\boldsymbol{\mu}_p, \boldsymbol{\Sigma}_p)$ and $q = \mathcal{N}(\boldsymbol{\mu}_q, \boldsymbol{\Sigma}_q)$, this integral resolves to the closed-form solution:

$$D_{KL}(p||q) = \frac{1}{2}\left[\log \frac{|\boldsymbol{\Sigma}_q|}{|\boldsymbol{\Sigma}_p|} + \text{tr}(\boldsymbol{\Sigma}_q^{-1}\boldsymbol{\Sigma}_p) + (\boldsymbol{\mu}_q - \boldsymbol{\mu}_p)^T\boldsymbol{\Sigma}_q^{-1}(\boldsymbol{\mu}_q - \boldsymbol{\mu}_p) - d\right]$$

We proceed by imposing the condition of scaled isotropic covariance, where $\boldsymbol{\Sigma}_p = \boldsymbol{\Sigma}_q = k\mathbf{I}$. This yields the following properties for the covariance matrix $\boldsymbol{\Sigma}_q$:

- Determinant: $|\boldsymbol{\Sigma}_q| = |k\mathbf{I}| = k^d$
- Inverse: $\boldsymbol{\Sigma}_q^{-1} = (k\mathbf{I})^{-1} = \frac{1}{k}\mathbf{I}$

Substituting these into the general formula:

$$D_{KL}(p||q) = \frac{1}{2}\left[\log \frac{k^d}{k^d} + \text{tr}\left(\left(\frac{1}{k}\mathbf{I}\right)(k\mathbf{I})\right) + (\boldsymbol{\mu}_q - \boldsymbol{\mu}_p)^T\left(\frac{1}{k}\mathbf{I}\right)(\boldsymbol{\mu}_q - \boldsymbol{\mu}_p) - d\right]$$

The expression simplifies term by term. The log-determinant ratio vanishes:

$$\log \frac{k^d}{k^d} = \log(1) = 0$$

The trace term becomes the dimension of the space, as the scalars cancel:

$$\text{tr}\left(\frac{1}{k} \cdot k \cdot \mathbf{I}\right) = \text{tr}(\mathbf{I}) = d$$

The quadratic form becomes the scaled squared Euclidean distance:

$$(\boldsymbol{\mu}_q - \boldsymbol{\mu}_p)^T\left(\frac{1}{k}\mathbf{I}\right)(\boldsymbol{\mu}_q - \boldsymbol{\mu}_p) = \frac{1}{k}(\boldsymbol{\mu}_q - \boldsymbol{\mu}_p)^T(\boldsymbol{\mu}_q - \boldsymbol{\mu}_p) = \frac{1}{k}||\boldsymbol{\mu}_q - \boldsymbol{\mu}_p||_2^2$$

Substituting these simplified components back into the equation, the dimensional terms cancel out:

$$D_{KL}(p||q) = \frac{1}{2}\left[0 + d + \frac{1}{k}||\boldsymbol{\mu}_q - \boldsymbol{\mu}_p||_2^2 - d\right]$$

This leaves us with the final, concise relationship:

$$D_{KL}(p||q) = \frac{1}{2k}||\boldsymbol{\mu}_p - \boldsymbol{\mu}_q||_2^2$$

This proves that under the assumption of shared, scaled isotropic covariance, the information-theoretic measure of KL divergence is mathematically equivalent to a simple, scaled Euclidean distance between the means of the distributions.

# B  TRAINING DETAILS

We fine-tuned a pre-trained Qwen2.5-0.5B decoder-only model on a mixed corpus drawn from WikiText-103, CNN/DailyMail, US-PD Books, Cosmopedia, SciQ, CodeParrot and TinyStories. Inputs were tokenized into fixed 1024-token blocks. Optimization used AdamW with $\beta_1 = 0.9$, $\beta_2 = 0.95$, $\varepsilon = 10^{-8}$ and weight decay 0.01, applying a linear warmup over the first 1% of steps followed by cosine decay. We scheduled the router inverse temperatures $\beta_{CE}$ and $\beta_{CU}$ from 0.1 to 100.0 using a cosine schedule with 100 warmup steps to progressively sharpen the gating decisions. Learning rates were set to $1 \times 10^{-5}$ for the backbone, $1 \times 10^{-3}$ for the PriorFFN or transition network, and $1 \times 10^{-2}$ for the predictive router. Training ran with a per-device batch size of 8 and gradient accumulation over 32 steps (effective batch size 256), using bfloat16 mixed precision and activation checkpointing. A fixed seed of 42 ensured reproducibility. The codebase is built on PyTorch (Paszke et al., 2019) and Hugging Face Transformers, configured via Hydra, distributed with Accelerate (FSDP), and batches prepared with the Hugging Face data collator. Final evaluation employed the lm-eval harness.