# OpenReview forum: "Subjective Depth and Timescale Transformers: Learning Where and When to Compute"
_ICLR.cc/2026/Conference — ICLR 2026 Conference Withdrawn Submission_

### Official Review · Reviewer_WzJK · 2025-10-14

**Soundness:** 3
**Presentation:** 3
**Contribution:** 3
**Rating:** 2
**Confidence:** 2

**Summary:**

The authors use Bayesian surprise to reduce redundant computation in decoder-only Transformers. This helps the model to learn to allocate computation only to tokens inducing a significant update in its internal representations, thereby providing a more effective inductive bias for routing than a generic importance score. The authors implement a Subjective Depth Transformer (SDT), that uses decision and dynamic layers. The decision layer generates a  full block output and a lightweight prediction of the output. The dynamic layer computes the mean squared error between these steps (as a tractable proxy for Bayesian surprise) to monitor expected change and unexpected changes. The Subjective Timescale Transformer (STT) consists of a lightweight Transition Network to predict the current token's residual update based on the previous token's processed state. This network is capable of handling dynamic capacities of load at test time. Both these methods use a unified surprise routing mechanism with a continuous gating score. The top-k tokens are chosen and a mixture of depth is applied when both of these models are used in unison. This reduces the need for computational capacity.

**Strengths:**

A> The idea of implementing Bayesian surprise is novel.

B> The two networks focus on improving two different aspects of efficiency.

C> The hardware aware design (modifying based on available memory) makes the pipeline versatile.

**Weaknesses:**

A> The optimal value of top-k is missing.

B> The experiments are performed on smaller LLM models. Comparison with existing Mixture of models that implement gating (example [1]) is missing.

C> The paper does not discuss any metrics that indicate computational load - for instance, MACs, FLOPs or number of trainable parameters.

[1] Huang, Haiyang, et al. "Toward efficient inference for mixture of experts." Advances in Neural Information Processing Systems 37 (2024): 84033-84059.

**Questions:**

The paper was an interesting read. I would like the authors to expand on the scalability of the method. The existing method works on a 0.5B model - is there a possibility for the pipeline to reduce computational load on larger models (that demonstrate superior performance)? Is it possible to extend the work to other transformer based tasks like ViT?

---

### Official Review · Reviewer_gR8A · 2025-10-31

**Soundness:** 1
**Presentation:** 3
**Contribution:** 1
**Rating:** 2
**Confidence:** 3

**Summary:**

The paper proposes using alternative metrics to supervise a router deciding whether or not to skip a layer for a token. Building upon the mixture of depths architecture which trains the router to predict the top-k mask, the authors suggest to instead train the router to predict a "surprise" metric. More specifically, they train a cheaper model to predict the difference between input and output of the transformer block. They propose two variants of this model, one depending only on the current token (SDT) and one depending on the previous token (STT). This prediction is then compared against the true difference between input and ouput of the transformer block and is used to compute a gate value on that block.

**Strengths:**

The paper's writing is mostly understandable and clear. The authors do a good job of explaining related works and establishing the relevance of this work in context of earlier works.  The general experimental setting is also sound (see below regarding caveats).

**Weaknesses:**

One of my major concerns is that the current method requires the block to be executed as the gating depends on the output of the block. This means that at inference time you still need to compute the block for all the tokens and then use the gating to decide whether or not to apply it. Therefore this only increases the amount of compute. In contrast, MoD leads to actual savings in compute at inference time as you can skip a block simply based on the router's decision.

Additionally, in light of the added compute, MoD is not the only suitable baseline here. There are earlier works that suggest adding gating to the residuals such as ReZero [1]. These would also be needed baselines to compare to.

On top of this, as pointed by the authors, the results are quite mixed and it is not clear whether there is any benefit to the proposed method. Particularly, several of the downstream tasks have around chance performance. I understand that obtaining confidence intervals can be compute intensive and therefore hard to obtain, but without them and given how close the current values are, it is really hard to make any interpretation of the effect of the proposed method.

Overall, while I think the idea is interesting, I think there are unresolved issues around both practicallity of its application and its efficacy in obtaining a good performance.

[1] Bachlechner, Thomas, Bodhisattwa Prasad Majumder, Henry Mao, Gary Cottrell, and Julian McAuley. "Rezero is all you need: Fast convergence at large depth." In Uncertainty in Artificial Intelligence, pp. 1352-1361. PMLR, 2021.

**Questions:**

1. Why didn't you explore dynamic capacity with SDT?

---

### Official Review · Reviewer_9Yaf · 2025-10-31

**Soundness:** 4
**Presentation:** 3
**Contribution:** 3
**Rating:** 4
**Confidence:** 5

**Summary:**

The authors introduces two conditional-compute variants for decoder-only Transformers: Subjective Depth Transformer (SDT) with alternating Decision/Dynamic layers, and Subjective Timescale Transformer (STT) with a unified temporal transition-based layer.​ They use an approximation to Bayesian surprise by treating hidden states as Gaussian means with shared isotropic covariance, yielding MSE between posterior and prior/transition predictions as a routing signal, combined via differentiable CE/CU criteria with a probabilistic OR and Top-K routing for fixed capacity.​ The authors provides a transfer-learning comparison (0.5B Qwen2.5 backbone) against a re-implemented Mixture-of-Depths (MoD) baseline at γ=0.5, with interleaving of custom layers, showing stable training and some cases where STT variants perform comparatively better among conditional models, though all lag the dense baseline on several standard benchmarks.

**Strengths:**

1. The surprise proxy grounded in a KL/MSE derivation gives a coherent and unified differentiable gate, aligning novelty (CU) and prediction (CE) criteria with predictive-coding literature and enabling direct optimization under LM loss via residual scaling.​
2. Fixed-capacity Top-K routing preserves a static compute graph, paralleling MoD while contributing a theoretically motivated score function and a causal router design suitable for autoregressive inference.
3. SDT separates prior formation from compute execution and enables analysis of spatial prior quality, while STT fuses them with a temporal transition network and optionally learns dynamic capacity via thresholds, broadening the conditional-compute design space.
4. Ablations on prior capacity and adaptation strategy (full FT vs LoRA) plus qualitative routing dynamics (shift from CU to CE over training, and reduced deeper-layer capacity in STT) provide useful insight into the mechanism’s behavior and inductive biases.
5. The paper is explicit about underperformance vs dense baselines in this constrained transfer setting and outlines future directions for scaling, capacity sweeps, alternative similarity metrics, and multi-seed significance.

**Weaknesses:**

1. The core routing signal assumes hidden states are draws from isotropic Gaussians with shared covariance, so $$ D_{\mathrm{KL}}\big(N(\mu_p,kI)\,\|\,N(\mu_q,kI)\big)=\frac{1}{2k}\|\mu_p-\mu_q\|_2^2 $$, which the method operationalizes as an MSE over residuals to define surprise $$D_{\mathrm{st}}$$ and $$D_{\mathrm{ch}}$$ after a 1/d rescaling, but the paper does not test the isotropy assumption nor report any calibration of this proxy against uncertainty or next-token error, leaving validity of the proxy unsubstantiated in transformer hidden spaces that are typically anisotropic and layer-normalized, not Gaussian. Concretely, the decision-layer SDT metrics and STT metrics use $$ D_{\mathrm{st},t}=\frac{1}{d}\|\Delta x_t\|_2^2 $$ and $$ D_{\mathrm{ch},t}=\frac{1}{d}\|\Delta x_t-\widehat{\Delta x}_t\|_2^2 $$, but there is no analysis comparing these to alternatives like layer-normalized cosine or entropy-like proxies, nor correlation plots with perplexity improvements, so the choice of $$ \ell_2 $$-MSE remains a heuristic despite the KL derivation under strong assumptions.

2. The hard predictive-coding criteria are softened as $$ \mathrm{CE}_t=D_{\mathrm{st},t}-(D_{\mathrm{ch},t}-\log o_{\mathrm{ce}}) $$ and $$ \mathrm{CU}_t=D_{\mathrm{st},t}-m_{\mathrm{cu}}\cdot \mathrm{MA}(D_{\mathrm{st},t}) $$, then combined via a probabilistic OR $$ g_{cont,t}=\sigma(\beta_{\mathrm{ce}}\mathrm{CE}_t)+\sigma(\beta_{\mathrm{cu}}\mathrm{CU}_t)-\sigma(\beta_{\mathrm{ce}}\mathrm{CE}_t)\sigma(\beta_{\mathrm{cu}}\mathrm{CU}_t) $$, but the paper does not establish this as a calibrated hypothesis test nor analyze sensitivity to the learnable inverse temperatures $$ \beta_{\mathrm{ce}},\beta_{\mathrm{cu}} $$, the moving-average horizon in CU, or the offset/multiplier, making the gate’s operating point potentially brittle and dataset-specific.

3. The paper analytically claims that with interleaving and capacity $$ \gamma $$, self-attention work reduces to $$ 0.5\cdot 1 + 0.5\cdot \gamma^2 $$ of the dense cost (e.g., $$ \gamma=0.5 \Rightarrow 62.5\% $$), but this relies on selecting both queries and keys/values to the same subset at routed layers, otherwise the cost scales like $$O(\gamma T\cdot T)$$ rather than $$O(\gamma^2 T^2)$$, and the paper does not clarify how keys/values from bypassed tokens are pruned or masked consistently with residual paths and cache correctness.

4. All conditional models consistently underperform the dense baseline on standard benchmarks in the 0.5B Qwen2.5 transfer setting, with several conditional scores near random-guessing on HellaSwag and ARC‑C, and there is no scaling, capacity sweep, or multi‑seed statistical analysis to test whether the approach improves with model/data scale or better hyperparameters.

5. The study fixes $$ \gamma=0.5 $$, interleaves every other layer, and evaluates a single small backbone with one seed, which is insufficient to substantiate broad claims about compute–accuracy trade‑offs or stability, especially when the practical promise is efficient long‑context generation.

**Questions:**

1. MoD-style update mixes routed and bypassed tokens $$ x_t^{(l)}=\mathbf{1}[t\in S^{(l)}]\cdot f^{(l)}(X_{S^{(l)}}^{(l-1)})_t + x_t^{(l-1)} $$, but at inference the SDT CR uses only $$x_{t}^{(l-1)}$$ while the teacher mask depends on pairwise token ordering and non-causal Top‑K across the full sequence, wouldn;t this create a distribution shift that is not measured? especially harmful when capacity is tight and mask errors propagate through depth?

2. It is unclear whether the added routing networks and scatter/gather operations negate the theoretical $\gemma^2$ attention savings?

---

### Official Review · Reviewer_GUuP · 2025-11-02

**Soundness:** 3
**Presentation:** 3
**Contribution:** 3
**Rating:** 4
**Confidence:** 3

**Summary:**

The paper proposes two conditional-compute architectures for decoder-only Transformers: Subjective Depth Transformer (SDT) and Subjective Timescale Transformer (STT). Both are driven by a surprise-based routing signal rather than a separate, semantically opaque router as in MoD. Concretely, the model computes a “posterior” representation and compares it either to the input (static surprise) or to a lightweight “prior” predictor (change surprise). The KL divergence under shared covariance is approximated by MSE, making the signal cheap and differentiable.

**Strengths:**

1. **Well-motivated signal**: Instead of training an extra router that learns arbitrary scores, the method derives the routing score from an interpretable discrepancy (posterior vs. prior / input), which is more principled and ties to predictive coding.

2. **Hardware-friendly design**: SDT keeps a static graph and fixed Top-K; STT also gives a fixed-capacity variant before introducing the dynamic one. This is practical.

3. **Training-dynamics insight**: they show CE and CU do not dominate at the same time; the signal shifts from novelty- to prediction-driven, which supports their framing.

**Weaknesses:**

1. **Main result is weak**: every conditional model is below the dense baseline, sometimes by a noticeable margin, so the narrative has to lean on “we saved compute.” But the paper reports mostly theoretical savings (e.g. 62.5% self-attention under γ=0.5) rather than real wall-clock / memory numbers on commodity GPUs.
2. **Single scale / single γ**: nearly all conclusions are drawn from Qwen2.5-0.5B with γ=0.5. To claim the method is robust, we need γ∈{0.25,0.5,0.75} or at least “every layer vs. every other layer.”
3. **KV-cache handling not fully specified**: STT claims to “manage” KV contribution, but the exact rule for unselected tokens (store or drop?) is not spelled out; this directly affects the claimed ~50% KV reduction.

**Questions:**

1. Can you report real inference speedups on A100/4090 for sequence lengths {32, 128, 1k} and a couple of batch sizes, including KV size changes?
2. Why is your MoD so low? Could you please either align your setup to the original MoD paper or explain the main departures (capacity, layer placement, optimizer, routing loss)?
3. **γ sweep**: even a small γ sweep on STT would help confirm that “temporal prior + surprise” is the real source of robustness, not a lucky γ.
If the authors can address the above issues (especially stronger MoD baselines and real inference savings), I am willing to raise my score.

---

### Note · Authors · 2025-12-02

**Comment:**

Dear Area Chair and Reviewers,

We are writing to formally withdraw our submission from consideration for ICLR 2026.

Following the feedback provided during the review process, particularly regarding the need for more rigorous baselines and scaling experiments, we conducted a more comprehensive evaluation of our proposed architectures, SDT and STT, against the MoD baseline.

While our new experiments at larger scales yielded some promising signals, the overall results remain inconsistent and merit further investigation to ensure robustness. Given these findings, we believe the work requires further refinement and analysis before it meets the conference's standards for publication.

We sincerely thank the reviewers (WzJK, gR8A, 9Yaf, GUuP) for their time and constructive comments, which were instrumental in identifying these issues and have guided our future direction for this research.

Sincerely,

The Authors

**Withdrawal Confirmation:**

I have read and agree with the venue's withdrawal policy on behalf of myself and my co-authors.